# Diet Quality and Eating Practices among Hispanic/Latino Men and Women: NHANES 2011–2016

**DOI:** 10.3390/ijerph18031302

**Published:** 2021-02-01

**Authors:** Francine Overcash, Marla Reicks

**Affiliations:** Department of Food Science and Nutrition 1, University of Minnesota, Minneapolis, MN 55410, USA; mreicks@umn.edu

**Keywords:** Hispanic/Latino adults, diet quality, men vs. women

## Abstract

Dietary quality of Hispanic/Latino adults residing in homes with children may differ by gender, that in turn, may impact youth through role modeling and food availability. Using a nationally representative sample (*n* = 1039) from the National Health and Nutrition Examination Survey (2011–2016), adjusted regression analyses were used to examine food-related practices, food group intake, and dietary quality among Hispanic/Latino men and women in homes with children (6–17 years). Compared to women, men had lower total 2015 Healthy Eating Index (HEI) scores and component HEI scores for healthy food groups. Men also ate more meals that were not home prepared/week and purchased more foods from non-grocery stores than women. Negative food-related practices and working more hours/week may explain in part the lower dietary quality observed among Hispanic/Latino men than women. Interventions may be improved by targeting gender-specific food-related behaviors that could positively impact dietary quality of youth residing with them.

## 1. Introduction

Improving diet quality to prevent obesity and reduce obesity-related disease risk factors among Hispanic/Latinos is a critical public health priority, given the Latino population is projected to increase from 18% of the total U.S. population to 29% by 2060 [1]. Obesity-related disease risk including risk of cardiovascular disease is substantial among Hispanic/Latinos in the U.S. with adverse levels of risk factors such as diabetes, hypertension and hypercholesterolemia [2,3,4]. Research suggests that diet quality among Hispanic/Latino adults is negatively influenced by sociodemographic and sociocultural factors such as younger age, less education, lower household income, and lower preference for using the Spanish language for reading, speaking, and listening [5].

Poor dietary quality and practices among Latino adults may have a strong impact on younger generations within the family unit, given long-held beliefs, traditions and sociocultural influences persistent in Latino culture. Compared to White American families, the basic Latino family unit is larger (e.g., inclusive of aunts, uncles, cousins, etc.) and more likely to reside in an intergenerational household [6]. The Latino value of *familismo*, loyalty, reciprocity, and solidarity within the immediate and extended family, exemplifies the belief that the family is the primary unit within Latino culture. As such, interdependence and influence regarding dietary practices over youth in the household may be extended to any adult living in a Latino household. 

Acculturation is a well-researched sociocultural factor impacting dietary patterns of immigrant populations. Greater acculturation has been associated with higher intakes of less healthy foods (sugar-sweetened beverages, fast food) and lower intakes of healthy foods (fruits, vegetables and whole grains) among Hispanic/Latino adults [7,8]. The influence of acculturation on dietary quality among Latino adults will also affect dietary quality of youth in the same households.

The significant and consistent associations between parents’ and children’s dietary intakes (e.g., intakes of total energy, fat, fruits, vegetables, and sugar-sweetened beverages) [9,10] underscore the importance of adults adhering to dietary guidance related to reduction of chronic disease risk [11]. A meta-analysis of 37 empirical studies examining parenting practices and child dietary intake found the strongest correlations between parental modeling and both healthy (r = 0.32, *p* < 0.001) and unhealthy child food consumption (r = 0.34, *p* < 0.001) [12]. Despite the evidence, few studies have focused on dietary parental modeling and child intake specific to Hispanic/Latino families with older children and adolescents. 

Differences in dietary intake and eating behaviors between Hispanic/Latino mothers and fathers may affect the manner in which parents model food and beverage consumption and make foods available in the home environment and away from home; however, limited studies have examined these differences. Understanding sociodemographic and sociocultural differences in diet quality and consumer behavior among Hispanic/Latino parents may provide better insight to inform family-centered interventions to prevent obesity. While mothers are traditionally considered the primary food planner and preparer in Hispanic/Latino homes, fathers are increasingly being recognized as having an important influence on youth diet quality and food choices based on food parenting practices such as modeling and making foods available [13]. Therefore, the purpose of this study was to provide a descriptive analysis of dietary intake and food-related practices among Latino adult men and women aged 25–50 who resided in a household with children (6–17 years) from National Health and Nutrition Examination Survey (NHANES) (2011–2016). Given these characteristics, adults were presumed to be a parent or guardian or extended family member who had the potential to influence dietary intake of children in the household. 

## 2. Materials and Methods

This cross-sectional analysis used data from the NHANES, an ongoing, public access population-based survey which employed a complex, stratified multistage probability sample design to create a representative sample of the non-institutionalized, civilian U.S. population [14]. Each survey consisted of questionnaires administered in the home by trained personnel followed by a standardized health examination that took place in a mobile examination center (MEC). The MEC session included a 24-h dietary recall. New survey data are released in 2-year cycles. NHANES data collection has evolved over the years, including oversampling of certain subgroups (e.g., Hispanics, Non-Hispanic black persons, Non-Hispanic Asians, persons over 80 years, etc.) in order to increase reliability and precision of certain populations of public health interest. NHANES was conducted according to guidelines set forth by the Declaration of Helsinki, and all procedures involving human subjects were approved by the National Center for Health Statistics Institutional Review Board for the Ethics Review Board (protocol no. 2011–17) [15]. Written informed consent was obtained from all participants before data collection. The University of Minnesota Institutional Review Board determined that analysis of these de-identified, publicly available data was not human research according to U.S. Department of Health and Human Services and Food and Drug Administration guidelines.

The final analytic sample consisted of *n* = 1039 adult participants from three two-year cycles of NHANES (2011–2016). Participants were included in the current study if they had self-reported Hispanic ethnicity, were aged 25–50, resided with a child between the ages of 6–17, and if their in-person dietary intake data were deemed reliable and meeting established criteria by the trained interviewer. 

Sociodemographic data were collected during the home interview. Body mass index (BMI, weight in kg/height in meters^2^) was calculated from body measurement data collected during the MEC visit by trained health technicians. BMI status was defined by the Centers for Disease Control: underweight (BMI < 18.5), normal or healthy weight (BMI 18.5–24.9), overweight (BMI 25.0–29.9), and obese (BMI ≥ 30.0). 

The 24-h dietary recall data collected during the MEC session was used to determine dietary intake and quality assessment in the current study. The trained interviewers conducted the recall in Spanish or English with translators available if needed employing the USDA’s Automated Multiple-Pass Method [16]. The Food Patterns Equivalents Databases (FPED) data used in the current study was based on NHANES dietary recall data from each cycle to calculate intake amounts of food groups (e.g., total fruit, total vegetable, whole grains, total dairy, etc.). Developed by the USDA, FPED are public use datasets that convert the foods and beverages in the Food and Nutrient Database for Dietary Studies to the 37 USDA Food Patterns components to evaluate food and beverage intakes of Americans with respect to the 2015–2020 Dietary Guidelines for Americans recommendations [17]. The food pattern components are defined as the number of cup equivalents of fruit, vegetables, and dairy; ounce equivalents of grains and protein foods; teaspoon equivalents of added sugars; gram equivalents of solid fats and oils; and number of alcoholic drinks [18].

The Healthy Eating Index (HEI) 2015 is the most current measure of dietary quality to assess how well dietary intake meets the recommendations set forth by the Dietary Guidelines for Americans 2015–2020 [18]. The HEI is density-based (e.g., amounts per 1000 kcal) rather than based on absolute amounts and relies on a common set of standards that are applicable across individuals and settings [19,20]. The HEI yields a total score, indicative of overall dietary quality (maximum score = 100). That is, the greater the total HEI score, the closer the overall diet aligns with the 2015–2020 Dietary Guidelines for Americans (DGA). The total HEI score is the sum of 13 component scores with either 5 or 10 points per component to equal 100: total intact or cut fruit (including fruit juice), total whole fruit, whole grains, dairy, total protein, seafood and plant protein, greens and beans, total vegetables, fatty acids, refined grains, sodium, added sugars, and saturated fats, that can be examined collectively to reveal a pattern of dietary quality in relation to multiple dietary dimensions. Higher scores for the following components reflect closer alignment to the DGA regarding adequacy: total intact or cut fruit, total whole fruit, whole grains, dairy, total protein, seafood and plant protein, greens and beans, total vegetables and fatty acids. Higher scores for the following components reflect closer alignment to the DGA regarding moderation: refined grains, sodium, added sugars, and saturated fats, and indicate lower intakes. Total HEI and the component HEI scores were calculated using the dietary recall data imported into a SAS^®^ program (version 9.4) [21] developed by National Institutes of Health-NCI, Division of Cancer Control & Population Studies [21].

Eating practices were assessed using four questions from the Diet Behavior and Nutrition questionnaire within NHANES that was completed during the home interview [14]. The questions pertained to frequencies and types of meals prepared away from home and use of convenience foods. Food-driven consumer behaviors were assessed using four questions from the Consumer Behavior questionnaire also completed during the home interview [14]. These questions asked the participant to estimate the amount of money spent at grocery stores, stores other than grocery stores/supermarkets and non-home prepared meals. The questions used in the current study were selected as those likely to influence diet quality based on previous studies and relevance to food-driven consumer behaviors [22,23,24,25,26].

Data analyses were conducted using SAS^®^ Survey Procedures with NHANES supplied sampling weights to account for the complex, multistate, probability sampling design of NHANES data. Sociodemographic characteristics were collapsed into categories after examination of the variable distributions. Differences by sex across categories of demographic and BMI status were determined by chi-square test using PROC SURVEYFREQ. Regression analyses (least square means and standard errors) using Wald’s F test of PROC SURVEYREG were used to determine differences by sex for the dietary variables (including HEI Scores), eating practices, and food-driven consumer behavior variables. Regression models were adjusted for marital status, marital status/sex interaction term, annual family income, BMI status, and energy (kcal). All reported *p*-values were two-tailed and considered statistically significant at *p* < 0.05. 

## 3. Results

### 3.1. Sociodemographic Characteristics and BMI Status

The final dataset consisted of more women (56%) than men (44%) (Table 1). Significantly more women than men were divorced/separated/never married (vs. married/living with partner), reported annual family income ≤ $34,999 (vs. ≥$35,000), were below the official definition of poverty (i.e., income-to-poverty ratio < 1.00), and were not currently working at a job or business. Only 17% and 14% of women and men, respectively, had weight status in the normal weight category. A greater percentage of women had a BMI in the obese category compared to men. For those who worked at a job or a business, men worked almost nine more hours in the past week compared to women. 

### 3.2. Energy-Adjusted Mean Dietary Intake and HEI Scores

Compared to men, women reported significantly higher mean total vegetable intake, and intakes from vegetable subgroups (dark green, red/orange, and starchy vegetables), and intake of total dairy foods (Table 2). Men consumed more total protein both inclusive and exclusive of legumes than women. Women attained a significantly higher total HEI score in comparison to men (Table 3). Similar differences between men and women were found in HEI component scores. For example, women had a higher mean total vegetable HEI score and mean total dairy HEI score than men. Likewise, men also had a higher mean score of total protein than women. Women had higher component HEI scores for total fruit and whole fruit than men (Table 3).

### 3.3. Food-Related Practices

Men reported eating almost 1.5 more meals in the past week that were not home prepared than women (mean_men_ = 3.9 vs. mean_women_ = 2.6) (Table 4). Meals from fast food or pizza place restaurants represented the majority (mean_men_ = 2.7) of these non-home+` prepared meals in the past 7 days for men; a mean that was also higher than the mean for women (mean_women_ = 2.1). The number of ready-to-eat meals or number of frozen meals/pizza in the past 30 days did not differ by sex. Men reported spending almost $30 more on foods at stores other than supermarkets or grocery stores in the past 30 days compared to women (Table 4). No other differences between sexes were found for any other food-driven consumer behavior measured.

## 4. Discussion

In this study, the results generally indicated that the energy-adjusted intake of healthier foods was greater among Hispanic/Latino women than men. Women had higher mean intakes of total vegetables, vegetable subgroups (dark green, red/orange, starchy), total dairy and lower intakes of total protein than men. These differences were reflected in the total HEI score, and component scores for total vegetable, total protein, total fruit, and whole fruit. In addition, women consumed fewer meals that were not home-prepared than men. These results reflect potential differences in how men (fathers) and women (mothers) may model dietary intake and make different types of foods available to children residing in the same households. Previous studies using NHANES data presumed that the household reference person was the parent or guardian of the child within the household [29] or that the child’s weight status and therefore, dietary behavior was influenced by family income and the head of the household’s education level [30]. Another approach is to presume that an adult of child-rearing age residing in a household with a child 6–17 years is the parent or guardian of the child and therefore influences the child’s diet.

The current study found more Latino men than women worked outside the home, which may explain the lower mean intakes of healthy foods and lower total and component HEI scores among men compared to women based on increased opportunities to eat meals outside the home. In other studies, away from home meals were of poorer nutritional quality compared to meals prepared from scratch at home [31,32,33,34]. A 2018 report from the USDA, Economic Research Service found employed individuals reported higher weekly frequencies of food away from home compared to those unemployed [35]. Intake of meals prepared outside the home and convenience foods or ‘ready-meals’ is prevalent among Hispanic/Latino adults and has been associated with poorer diet quality and health outcomes [36,37]. 

Previous qualitative studies among Latino men found work demands made convenience and taste a priority over more healthful food choices, despite an awareness of the importance of healthy eating [38,39]. The strenuous nature, and therefore, high energy needs of some jobs may also make these types of foods more appealing [39]. In the current study, men spent more money on food than women at stores other than grocery stores or supermarkets. A possible explanation is that men may make more quick trips for foods to eat during the workday for meals or breaks away from home. 

Latino men have been shown to hold present-day orientation, a tendency to not think about the future and rather make choices that affect the immediate [40]. Researchers can use this cultural understanding to design interventions that are tailored to this need for immediacy. Family-centered interventions are generally considered to have a high potential for effectiveness in obesity prevention and treatment [41,42]. These interventions could inform fathers of healthier options being offered at restaurant chains, a growing movement in the industry [43] or of healthy alternatives available at stores other than grocery stores or convenience stores. Alternate modes of delivery for lessons may also be worth considering to better cater to demanding schedules and the need for immediacy.

Cultural norms may also influence dietary quality among Latino parents and explain differences between men and women. Compared with other ethnic groups, many Hispanic/Latino families tend to have traditional views regarding gender, including distinct gender role expectations for men and women [44,45]. Previous work indicated that Hispanic/Latino men have been socialized to follow the same sex-defined patterns established by previous generations, including a greater expectation for women to eat fruits and vegetables while men eat more meat and fats [39]. Based on a focus group study with Latino fathers, Zhang et al. [13] identified traditional gender roles as a socio-environmental factor that challenged father involvement in promoting healthy dietary intake of youth. Women were perceived as having primary responsibility for child rearing and food parenting practices, while men were perceived as lacking interest in nutrition or food preparation. A reliance on their female counterpart may result in Latino men having less motivation or knowledge to make healthier dietary choices. The current study’s results show fruit and vegetable intake needs to be improved among Latino men. Interventions aimed at Latino fathers should specifically promote involvement in fruit and vegetable procurement and preparation to enhance liking.

Fruits, vegetables, and dairy food groups contain components associated with decreased risk of type 2 diabetes [46]. Hispanic/Latino adults are 1.7 times more likely to have been diagnosed with diabetes and 1.4 times more likely to die from diabetes than other groups [47]. Based on the results of the current study, Hispanic/Latino women may have greater protection from this chronic disease than men because of higher HEI component scores for these healthy foods. Intake of dark green and red/orange vegetables were also higher among women than men, indicating that women may consume a wider variety of vegetables, which has been linked to greater amounts consumed [48]. The current study’s dietary findings can be considered together with other differential factors between Hispanic men vs. women shown to contribute to type 2 diabetes risk including sedentary and physical activity levels, chronic stress, and smoking [49] in designing more comprehensive diabetes prevention interventions for this population.

The current study’s results suggest Hispanic/Latino women may also serve as stronger role models than men for intake of fruit, vegetables, and dairy foods by youth. Role modeling has been shown to be an effective parenting practice related to promotion of healthy food intakes by youth [12]. Given that Hispanic/Latino families are characterized by strong family connections [50,51] and sharing meals together at home [52,53], dietary role modeling may have stronger impact for Hispanic/Latino youth compared to other groups. U.S. Hispanic/Latino adolescents had total HEI scores of 53.8/100 indicating a need for improvement with scores of 2.3/5 for fruit, 2.0/5 for vegetables, 4.5/10 for whole grains, and 7.4/10 for dairy based on a cross-sectional population-based cohort study from 4 U.S. communities [54]. The need for improvement is not based only on cross-sectional data. One focus group study found Hispanic/Latino mothers believed their husbands hindered healthy dietary patterns in the home by bringing home high-calorie foods for the family to eat based on their own preferences [55]. The current study lends further support to the emerging research area of the paternal role and influence Hispanic fathers play within the family’s home food environment [13]. Interventions aimed at improving intake of these healthy foods among Hispanic/Latino men and women residing in homes with children will also increase frequency of role modeling intake for youth. 

### Limitations

This study has several limitations. Dietary intakes and HEI scores were based on self-report, which may be prone to inaccuracy related to recall and portion size estimation errors. Lastly, the data were collected on a cross-sectional basis at one time point, limiting the ability to examine trends in intake and diet quality over time.

## 5. Conclusions

Overall diet quality based on total HEI scores were in need of improvement for men and women residing in homes with children 6–17 years. However, compared to Hispanic/Latino women, men had lower overall dietary quality and component dietary quality scores based on intake of healthy food groups, whereas a greater proportion of women had a BMI in the obese category compared to men. Hispanic/Latino men ate more meals not prepared at home and spent more money on food at stores other than grocery stores or supermarkets than women. Both practices could partially explain the differences in diet quality between men and women. 

## Figures and Tables

**Table 1 ijerph-18-01302-t001:** Sociodemographic characteristics and BMI (Body mass index) status for women and men.

Characteristic	Women (*n* = 582)		Male (*n* = 457)		
	Freq	%	Freq	%	*p*-Value ^a^
Age					
25–29	70	13.5	54	13.2	0.97
30–40	292	53.4	237	53.4	
41–50	220	33.1	166	33.4	
Education level					
≤High school/GED	260	43.7	217	46.3	0.42
Some college/college Grad/graduate school	322	56.3	240	53.7	
Marital status					
Married/living with partner	422	73.8	399	87.9	**<0.0001**
Divorced/separated/never married	155	26.2	54	12.1	
Family income level ^b^					
≤$34,999	255	49.6	164	41.4	**<0.0001**
≥$35,000	263	50.4	237	58.6	
Poverty ratio level ^c^					
Below official poverty definition	202	38.3	127	31.1	**0.0003**
Above official poverty definition	316	61.7	274	68.9	
Primary language spoken at home					
Only Spanish OR more Spanish than English	318	54.4	277	58.3	0.16
Equal Spanish/English, more English than Spanish, only English	262	45.6	180	41.7	
Country of birth					
Born in 50 US states or Wash DC	184	32.0	127	29.2	0.14
Other	398	68.0	328	70.8	
Citizenship status					
US citizen	286	50.7	198	46.2	0.10
Non-US citizen	291	49.3	252	53.8	
No. of years in the US					
<15 years	160	44.8	126	43.5	0.72
≥15 years	210	55.2	169	56.5	
BMI status					
Normal	99	17.3	63	13.6	**0.03**
Overweight	184	32.3	187	40.9	
Obese	293	50.4	202	45.5	
Employment status ^d,e^					
Working at a job or business	333	56.9	397	86.6	**<0.001**
With a job or business but not at work	11	1.8	10	2.5	
Looking for work/not working at a job or business	238	41.3	49	10.9	
Mean number of hours worked at all jobs/business in the last week ^e^	Mean	SE	Mean	SE	
	36.9	0.28	45.9	0.29	**<0.001**

^a^*p*-value for chi-square test; significant *p* < 0.05-bolded; ^b^ For reference, the mean of the median family income among Hispanic/Latino private households in the U.S for the years 2011–2016 was $47,609 [27]; ^c^ Defined using the Income-to-poverty ratio. Ratios below 1.00 indicate that the income for the respective family or unrelated individual is below the official definition of poverty, while a ratio of 1.00 or greater indicates income above the poverty level [28]; ^d^ Multiple choice question (Q150) on Occupation Questionnaire: “The next questions are about your current job or business. Which of the following {were you/was} doing last week?”. ^e^ Only applies to those who answered Yes to working at a job or business.

**Table 2 ijerph-18-01302-t002:** Least square means of food groups and subgroups for women and men ^a^.

Food Group	Women		Men		
	LS Mean	SE	LS Mean	SE	*p*-Value ^b^
Total intact or cut fruits and fruit juices (cup eq.)	1.14	0.08	0.92	0.16	0.07
Total vegetables (cup eq)	1.54	0.15	1.31	0.20	**0.004**
Dark green vegetables (cup eq.)	0.09	0.02	0.05	0.02	**0.03**
Total red and orange vegetables (tomatoes + other red and orange) (cup eq.)	0.45	0.05	0.37	0.07	**0.01**
White potatoes (cup eq.)	0.27	0.03	0.21	0.05	0.08
Total starchy vegetables (white potatoes + other starchy) (cup eq.)	0.37	0.04	0.29	0.06	**0.03**
Other vegetables not in the vegetable components listed above (cup eq.)	0.63	0.11	0.60	0.14	0.58
Legumes computed as vegetables (cup eq.)	0.25	0.04	0.27	0.04	0.56
Whole grains (oz. eq.)	0.58	0.09	0.54	0.12	0.60
Refined or non-whole grains (oz. eq.)	7.66	0.22	7.83	0.27	0.42
Total whole and refined grains (oz. eq.)	8.24	0.21	8.38	0.23	0.49
Total meat, poultry, seafood, organ meats, and cured meat (oz. eq.)	4.63	0.17	6.16	0.23	**<** **0.01**
Total meat, poultry, seafood, organ meats, cured meat, eggs, soy, and nuts and seeds; excludes legumes (oz. eq.)	5.91	0.21	7.38	0.25	**<** **0.01**
Total milk, yogurt, cheese, and whey (cup eq.)	1.74	0.06	1.41	0.06	**<** **0.01**
Foods defined as added sugars (tsp. eq.)	18.46	0.66	18.25	0.65	0.79

^a^ Final model co-variates: marital status marital status/sex interaction term, annual family income, BMI status, energy (kcal); ^b^
*p*-value significance: <0.05-bolded.

**Table 3 ijerph-18-01302-t003:** Healthy eating index 2015: Total HEI (Healthy Eating Index) score and component scores for women and men ^a^.

Score	Maximum Points	Women		Men		
		Mean	SE	Mean	SE	*p*-Value ^b^
Total HEI score	100	51.00	0.62	48.17	0.74	<0.01
Component scores						
Total vegetable	5	3.13	0.13	2.85	0.14	**<0.01**
Greens and bean	5	2.05	0.19	1.81	0.17	0.06
Total fruit	5	2.35	0.13	1.98	0.25	**0.04**
Whole fruit	5	2.35	0.16	1.86	0.28	**0.01**
Whole grain	10	1.83	0.23	1.51	0.30	0.09
Refined grains	10	4.59	0.21	4.39	0.25	0.36
Total dairy	10	5.35	0.12	4.50	0.16	**<** **0.01**
Total protein	5	4.23	0.10	4.52	0.10	**<** **0.01**
Seafood and plant protein	5	2.69	0.19	2.48	0.20	0.11
Fatty acids	10	5.00	0.18	4.95	0.20	0.84
Sodium	10	4.64	0.16	4.25	0.15	0.10
Saturated fats	10	6.05	0.15	6.45	0.18	0.09
Added sugars	10	6.69	0.17	6.62	0.14	0.74

^a^ Final model co-variates: marital status marital status/sex interaction term, annual family income, BMI status, energy (kcal); ^b^
*p*-value significance: <0.05-bolded.

**Table 4 ijerph-18-01302-t004:** Food-related practices among Hispanic/Latino women and men ^a^.

Eating Practices	Women		Men		
	Mean	SE	Mean	SE	*p*-Value ^b^
No. meals not home prepared past week	2.63	0.16	3.86	0.20	**<0.001**
No. meals from fast food or pizza place past 7 days	2.07	0.13	2.70	0.21	**0.01**
No. ready-to-eat foods past 30 days	1.91	0.29	2.59	0.52	0.26
No. frozen meals/pizza past 30 days	1.48	0.20	1.59	0.26	0.74
**Food-driven Consumer Behaviors**	**Mean $ Amount**	**SE**	**Mean $ Amount**	**SE**	
During the past 30 days, how much money {did your family/did you} spend at supermarkets or grocery stores? Please include purchases made with food stamps.	584.37	16.15	564.79	22.91	0.41
About how much money {did your family/did you} spend on food at other than grocery and supermarkets?	132.82	13.02	160.78	20.18	**0.04**
During the past 30 days, how much money {did your family/did you} spend on eating out? Please include money spent in cafeterias at work or at school or on vending machines, for all family members.	160.88	7.86	183.84	14.30	0.08
During the past 30 days, how much money {did your family/did you} spend on food carried out or delivered? Please do not include money you have already told me about.	25.90	4.30	24.34	3.12	0.62

^a^ Final model co-variates: marital status marital status/sex interaction term, annual family income, BMI status, energy (kcal); ^b^
*p*-value significance: <0.05-bolded.

## Data Availability

Data available in a publicly accessible repository. The data presented in this study are openly available https://www.cdc.gov/nchs/nhanes/index.htm.

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
