# Peer review of "Diet Quality and Eating Practices among Hispanic/Latino Men and Women: NHANES 2011–2016"

_ijerph, 2021, doi:10.3390/ijerph18031302_

Round 1
Reviewer 1 Report
Dear Authors,
It is well written manuscript, however I hope you find the following commentary constructive to strengthen it:
Abstract:
Some important information is lacking: where the survey was conducted and how many participants were involved. Giving some numbers would be also more informative for the reader.
Introduction:
What do you mean by "lower Spanish language preference" (lines 29-30). Please explain.
line 45: Please explain abbreviations: FV, SSBs
General comments: The reader would benefit from expansion of the introduction.
What percentage of the population are Hispanic/Lation in US? Please add this information to the introduction.
Are there any data on the percentage of Latin American families where a male is the nutrition decision maker? It seems that with the traditional family model such a situation is very rare.
Materials and Methods
Please explain on what basis "person dietary intake data were deemed reliable" (lines 84-85).
line 86: Abbreviation BMI should be first explain in the text.
Results:
I do not understand the relevance of some of the characteristics of the characteristics, e.g. country of birth and citizenship status. Why are these criteria important in characterizing the population? Are these factors relevant to the discussion of the results? If not, why not skip them?
Table 4 needs some formatting? Significant differences should be marked (refers to all tables)
Discussion:
lines 197-199: This part is NOT supported by results presented.
General remark: In the discussion, the authors could relate their results more to the results of other authors to present data in a broader context.
Conclusions:
I do not understand the reference to the role of the male in modeling the habits / diet of children in household. It was not the subject of the study, and there is no information whether the surveyed men were decision makers in the food choices in the family. I consider that this conclusion is not supported by the results obtained.
Author Response
Please see the attachement.
Reviewer 2 Report
Comments to the paper: ijerph-1054432 Diet quality and eating practices among
Hispanic/Latino men and women: NHANES 2011- 2016.
I applaud the authors' interest in the subject of acculturation and their relationship to the quality of the diet. I have some comments to the paper:
Lines 102 – 106: It is not clear how authors calculated the diet-quality score in this analysis. Based on the HEI 2015, it is necessary to describe specifically how co-authors considered each item (food group). The 13 components in the original score (each of the calculation elements), can be interpreted in different ways and give the score different uses. It will be very useful to specify
whether or not there were variants in the calculation of the score.
Lines 110 – 111: The Diet Behavior and Nutrition questionnaire: the questionnaire used is not cited, although the use of the score is known, it would be very useful co-authors refer to it for easy identification. Also, explain why four questions were selected and what they were.
In table 2: Why co-authors considered fruit juices in the same food group for intact or cut fruits? It is well known that fruit juices have provide important amounts of sugars, hence they are considered in a high sugar food group. Although, the HEI 2015 include them as a fruit, it is questionable.
The grouping of foods into some categories is unclear with regard to their nutritional characteristics
or components. For example: total meat appears to be different food item than the each one of the
meat types. Were there considered sausages in the cured meat item? Soy is a food that can be considered as a vegetable (fresh), a legume or like beverage. Nuts and seed could be in a separate food group due to its nutritional components.
It will be very useful co-authors explain the most convenient reasons for food classification for use in the diet quality score. Authors could compare between 2015 HEI and 2020 HEI for a better utility of these data.
In table 3: It would be advisable for co-authors to discuss more broadly how each item of the score was computed, especially in terms of differences in total dairy and total protein consumption. Also with the components like sodium, saturated fats and sugars, since no nutrients but food groups were described.
Line 143: In the subtitle “Energy-adjusted mean dietary intake and HEI scores” it would be expectable to read about energy-adjusted method used for this analysis. It is not clear. Authors repeated in lines 172 and 173 energy-adjusted intake.
In this analysis co-authors did not include descriptions about percentage of energy intake in participants although is a useful component in diet quality.
It is unclear how the score was used in each of the dietary components. When the HEI indicates that from a certain amount of food consumed, a score is given, this study does not mention how consumption was evaluate.
It was not mentioned if HEI components, point values and standards for scoring were used for co authors in this paper.
In general, it should be reviewed the uniformity of the text format.
Reviewer 3 Report
This manuscript was well written. However, the authors should address the following comments about the study. 1. Please explain the abbreviations; Line 45 ‘FV, SSBs’. 2. All participants live with their child, but author did not analysis the influence on children’s eating behavior. I do not understand that why authors mention about ‘positively impact dietary quality of youth’ (line 19-20) and ‘men would allow for more positive role modeling of intakes and availability of healthy foods for children in the same household’(line 247-249).Author Response
Please see the attachment.

Round 2
Reviewer 1 Report
Dear Authors,
I appreciate the corrections and clarifications introduced.
Below are some minor technical notes to consider:
Table 1.
Family income. I would suggest to add (eg below the table) information with a minimum or middle income for the possibility of reference to the values adopted in the study for readers outside the US
Table 4.
"Mean $ Amt" please explain the abreviation?
I would suggest to separate Strengts and limitations from the Dscussion main text (in subsection).
Author Response
Dear Reviewer 1,
Thank you for your review. We have addressed your additional concerns in bold and italics below.
Family income. I would suggest to add (eg below the table) information with a minimum or middle income for the possibility of reference to the values adopted in the study for readers outside the US.
Because the family income variable for NHANES is a categorical variable, we were unable to run a simple mean. However, we made the following revision that we hope addresses you concern:
We have added the following footnote to the Table 1 (and corresponding reference to Reference list)
For reference, the mean of the median family income among Hispanic/Latino private household in the U.S for the years 2011-2016 was $ 47,609.
Table 4.
"Mean $ Amt" please explain the abreviation?
“Amt” has been revised to “Amount”
I would suggest to separate Strengts and limitations from the Dscussion main text (in subsection).
We have created a separate paragraph subsection called Limitations
Thank you again. Please let us know if you have any further revisions/questions. Thank you.
Kind regards,
Francine M. Overcash, PhD, MPH